# OpenReview forum: "Mitigating Watermark Forgery in Generative Models via Randomized Key Selection"
_ICLR.cc/2026/Conference — Submitted to ICLR 2026_

### Official Review · Reviewer_h8s7 · 2025-10-19

**Soundness:** 3
**Presentation:** 3
**Contribution:** 2
**Rating:** 6
**Confidence:** 3

**Summary:**

The authors proposes a defense against watermark forgery attacks in generative AI systems. It introduces a randomized key selection mechanism, where each generation uses a randomly sampled watermark key from a pool, and authenticity is confirmed only if exactly one key detects a watermark. This approachs are provably forgery-resistant even when attackers have access to many watermarked samples, provided they cannot distinguish which key was used. The method preserves model utility and is modality-agnostic, applying to both text and image generation. Empirical results show a reduction in forgery success rates from nearly 100% to as low as 2%, with negligible computational overhead. The authors also provides theoretical bounds, extensive experiments against adaptive and blind attackers, and an open-source implementation, demonstrating the method’s robustness and practicality

**Strengths:**

This paper demonstrates strong originality by reframing watermark forgery prevention as a key-randomization problem, introducing a simple yet powerful defense that does not rely on increasing watermark density or retraining models. Its theoretical contribution, which can be a provable bound on forgery success independent of sample size, representing a conceptual advance over prior multi-key and statistical approaches. The work is of high quality, combining rigorous analysis with extensive empirical validation across text and image modalities, including comparisons to adaptive and blind attackers. I the presentation clarity is systematic, with clear algorithmic descriptions (e.g., Algorithm 1), theoretical assumptions, and visualizations that make the defense mechanism easy to follow. The paper’s significance lies in its practical applicability for securing generative AI outputs, as it provides a low-overhead, easily deployable solution that enhances trust and accountability in content attribution systems, addressing an increasingly critical issue for AI providers. The exposition is clear and well-structured, with intuitive figures (e.g., Figure 1) that help visualize the threat model and defense workflow. The writing maintains a good balance between formalism and accessibility—technical sections are precise, while the main ideas remain easy to follow. The inclusion of algorithms, assumptions, and calibration procedures aids reproducibility.

**Weaknesses:**

The problem is a bit of artificial -- foragaing the content to hurt the reputation of model providers, which can be less important than the standard settings.
The adaptive attacks are implemented as either a blind adaptive (averaging) attacker or a stronger informed/key-classification attacker, suggesting. This rests on the key assumption that attacker must obtain either many unlabeled samples (for averaging) or many labeled samples per key (for the informed attack), and the defender’s detection API is assumed inaccessible to the attacker (private detector). Moreover, a better training methodology for distinguishing watermarked content with different keys can be used for designing adaptive attack, such as using reward models trained via Bradley-Terry loss as in Watermarks In the Sand paper.
It is better to add latest models like Qwen3, Gemma2 and Llama3 if possible. In general the experiments can be strengthened in various ways.
Figure 7 has image captions that are overlapped and it is hard to interpret the results.

**Questions:**

I understand the setting can be new but is there any baseline you can add to the plot such as the one using reward models or recursive paraphrasing?

---

> ### Author Response · Authors · 2025-11-23
> **Official Response by Authors to Reviewer h8s7 (1/3)**
>
> We sincerely thank the reviewer for the detailed and positive feedback highlighting the paper’s "strong originality", "rigorous analysis", and "clear and well-structured" presentation. Please find our responses to all comments below.
>
> ---
>
> > The problem is a bit of artificial — forging the content to hurt the reputation of model providers, which can be less important than the standard settings
>
> We appreciate the reviewer’s perspective on the hierarchy of threat models. While we agree that watermark removal (evasion) is the classical setting, we argue that forgery (spoofing) has emerged as an equally critical "Day 0" threat due to recent legislative and industrial shifts.
>
> As noted in the Introduction (Section 1), upcoming regulations (e.g., EU AI Act, California AI Disclosure Bill) mandate technical provenance for AI content. If a watermark can be forged, malicious actors can frame compliant providers (like Google or OpenAI) for generating illegal, hateful, or disinformation content. This transforms forgery from a "reputation" issue into a potential legal and liability risk for model providers. With the public deployment of schemes like SynthID from Google Deepmind this is no longer a hypothetical threat. Prior work [A] has demonstrated that existing heuristical schemes are vulnerable to these spoofing attacks. Our work provides a defense that resists forgery attacks with minimal implementation overhead and small computational cost.
>
> ---
>
> > [...] a better training methodology for distinguishing watermarked content with different keys can be used for designing adaptive attack, such as using reward models trained via Bradley-Terry loss as in Watermarks In the Sand paper.
>
> We thank the reviewer for this insightful suggestion. We agree that advanced training methods, such as the Bradley-Terry reward models used in the "Watermarks in the Sand" paper, could potentially improve an adaptive attacker's ability to distinguish between keys.
>
> Rather than evaluating one specific classifier, we chose to address the reviewer's concern by performing a sensitivity analysis that bounds the performance of any classifier. We modeled the dependency between Key-Classification Accuracy and Forgery Success, treating classification errors as i.i.d.
>
> The table below (which we will add to the revision) demonstrates the forgery success rate across the full spectrum of classifier capabilities, from random guessing (0.0) to a perfect oracle (1.0).
>
> | Classification Accuracy | 0.0 | 0.3 | 0.5  | 0.7  | 0.9  | 1.0  |
> | ----------------------- | --- | --- | ---- | ---- | ---- | ---- |
> | Forgery Success (%)     | 0.0 | 6.0 | 46.0 | 58.0 | 65.0 | 78.0 |
>
> Our analysis confirms that our defense degrades gracefully. Even if a sophisticated attacker (using Bradley-Terry loss) achieves extremely high accuracy (e.g., 90%), the forgery success rate remains capped.
>
> We will include these experiments in our revised paper.

---

> > ### Author Response · Authors · 2025-11-23
> > **Official Response by Authors to Reviewer h8s7 (2/3)**
> >
> > > It is better to add latest models like Qwen3, Gemma2 and Llama3 if possible. In general the experiments can be strengthened in various ways.
> >
> > We appreciate the reviewer’s suggestion to include results for newer open-weight models. In the main paper, we evaluated forgery resistance using Mistral-7B as the provider model, where attackers had direct model access. To strengthen our experimental coverage, we additionally ran experiments where attackers had surrogate access to Gemma-2B, Gemma-7B, and Llama-7B. These models reflect a broader spectrum of open and instruction-tuned architectures and allow us to verify that our proposed randomized-key method consistently improves forgery resistance across model families and sizes.
> >
> > In summary, across all evaluated models, forgery success decreases consistently with the number of keys, and our randomized-key method provides the largest performance gain while preserving quality, mirroring the trends reported in the main paper.
> >
> > Below, we show the results for KGW-SelfHash and Unigram watermarking under both AdvBench and RealHarmfulQ datasets. "Base" refers to the single-key watermarking baseline; "Kirchenbauer" is the multi-key baseline (Kirchenbauer et al., 2023); and "Ours" refers to our randomized key selection approach.
> >
> > 1. Mistral-7B (from main paper) with Kirchenbauer's baseline included
> >
> > |     #Keys $r$    | Base (AdvBench) | Kirchenbauer |   Ours   | Base (RealHarmfulQ) | Kirchenbauer |   Ours   |
> > | :--------------: | :-------------: | :----------: | :------: | :-----------------: | :----------: | :------: |
> > | **KGW-SelfHash** |                 |              |          |                     |              |          |
> > |         1        |       0.75      |     0.75     |   0.75   |         0.67        |     0.67     |   0.67   |
> > |         2        |       0.75      |     0.68     | **0.53** |         0.67        |     0.64     | **0.49** |
> > |         3        |       0.75      |     0.56     | **0.33** |         0.67        |     0.56     | **0.31** |
> > |         4        |       0.75      |     0.50     | **0.26** |         0.67        |     0.42     | **0.23** |
> >
> > |  #Keys $r$  | Base (AdvBench) | Kirchenbauer |   Ours   | Base (RealHarmfulQ) | Kirchenbauer |   Ours   |
> > | :---------: | :-------------: | :----------: | :------: | :-----------------: | :----------: | :------: |
> > | **Unigram** |                 |              |          |                     |              |          |
> > |      1      |       0.77      |     0.77     |   0.77   |         0.64        |     0.64     |   0.64   |
> > |      2      |       0.77      |     0.73     | **0.56** |         0.64        |     0.62     | **0.49** |
> > |      3      |       0.77      |     0.70     | **0.12** |         0.64        |     0.55     | **0.14** |
> > |      4      |       0.77      |     0.68     | **0.16** |         0.64        |     0.56     | **0.16** |
> >
> >
> > 2. Gemma-2B (Attacker has surrogate access)
> >
> > |     #Keys $r$    | Base (AdvBench) | Kirchenbauer |   Ours   | Base (RealHarmfulQ) | Kirchenbauer |   Ours   |
> > | :--------------: | :-------------: | :----------: | :------: | :-----------------: | :----------: | :------: |
> > | **KGW-SelfHash** |                 |              |          |                     |              |          |
> > |         1        |       0.71      |     0.71     |   0.71   |         0.65        |     0.65     |   0.65   |
> > |         2        |       0.71      |     0.63     | **0.40** |         0.65        |     0.61     | **0.36** |
> > |         3        |       0.71      |     0.52     | **0.21** |         0.65        |     0.52     | **0.19** |
> > |         4        |       0.71      |     0.45     | **0.14** |         0.65        |     0.41     | **0.11** |
> >
> > |  #Keys $r$  | Base (AdvBench) | Kirchenbauer |   Ours   | Base (RealHarmfulQ) | Kirchenbauer |   Ours   |
> > | :---------: | :-------------: | :----------: | :------: | :-----------------: | :----------: | :------: |
> > | **Unigram** |                 |              |          |                     |              |          |
> > |      1      |       0.72      |     0.72     |   0.72   |         0.63        |     0.63     |   0.63   |
> > |      2      |       0.72      |     0.70     | **0.44** |         0.63        |     0.61     | **0.36** |
> > |      3      |       0.72      |     0.69     | **0.08** |         0.63        |     0.54     | **0.05** |
> > |      4      |       0.72      |     0.66     | **0.05** |         0.63        |     0.55     | **0.03** |

---

> > > ### Author Response · Authors · 2025-11-23
> > > **Official Response by Authors to Reviewer h8s7 (3/3)**
> > >
> > > 3. Gemma-7B
> > >
> > > |     #Keys $r$    | Base (AdvBench) | Kirchenbauer |   Ours   | Base (RealHarmfulQ) | Kirchenbauer |   Ours   |
> > > | :--------------: | :-------------: | :----------: | :------: | :-----------------: | :----------: | :------: |
> > > | **KGW-SelfHash** |                 |              |          |                     |              |          |
> > > |         1        |       0.73      |     0.73     |   0.73   |         0.66        |     0.66     |   0.66   |
> > > |         2        |       0.73      |     0.65     | **0.48** |         0.66        |     0.61     | **0.45** |
> > > |         3        |       0.73      |     0.58     | **0.29** |         0.66        |     0.52     | **0.27** |
> > > |         4        |       0.73      |     0.42     | **0.22** |         0.66        |     0.41     | **0.19** |
> > >
> > > |  #Keys $r$  | Base (AdvBench) | Kirchenbauer |   Ours   | Base (RealHarmfulQ) | Kirchenbauer |   Ours   |
> > > | :---------: | :-------------: | :----------: | :------: | :-----------------: | :----------: | :------: |
> > > | **Unigram** |                 |              |          |                     |              |          |
> > > |      1      |       0.77      |     0.77     |   0.77   |         0.62        |     0.62     |   0.62   |
> > > |      2      |       0.77      |     0.73     | **0.51** |         0.62        |     0.61     | **0.43** |
> > > |      3      |       0.77      |     0.70     | **0.15** |         0.62        |     0.56     | **0.13** |
> > > |      4      |       0.77      |     0.67     | **0.13** |         0.62        |     0.55     | **0.11** |
> > >
> > > 4. Llama-7B
> > >
> > > |     #Keys $r$    | Base (AdvBench) | Kirchenbauer |   Ours   | Base (RealHarmfulQ) | Kirchenbauer |   Ours   |
> > > | :--------------: | :-------------: | :----------: | :------: | :-----------------: | :----------: | :------: |
> > > | **KGW-SelfHash** |                 |              |          |                     |              |          |
> > > |         1        |       0.78      |     0.78     |   0.78   |         0.70        |     0.70     |   0.70   |
> > > |         2        |       0.78      |     0.68     | **0.47** |         0.70        |     0.63     | **0.44** |
> > > |         3        |       0.78      |     0.56     | **0.28** |         0.70        |     0.55     | **0.26** |
> > > |         4        |       0.78      |     0.49     | **0.20** |         0.70        |     0.41     | **0.18** |
> > >
> > > |  #Keys $r$  | Base (AdvBench) | Kirchenbauer |   Ours   | Base (RealHarmfulQ) | Kirchenbauer |   Ours   |
> > > | :---------: | :-------------: | :----------: | :------: | :-----------------: | :----------: | :------: |
> > > | **Unigram** |                 |              |          |                     |              |          |
> > > |      1      |       0.77      |     0.77     |   0.77   |         0.64        |     0.64     |   0.64   |
> > > |      2      |       0.77      |     0.75     | **0.49** |         0.64        |     0.57     | **0.42** |
> > > |      3      |       0.77      |     0.67     | **0.14** |         0.64        |     0.50     | **0.12** |
> > > |      4      |       0.77      |     0.61     | **0.12** |         0.64        |     0.50     | **0.10** |
> > >
> > > Across all models and datasets, our randomized-key “exactly-one” detection rule consistently yields the largest reduction in forgery success compared to both single-key and prior multi-key baseline, even when attackers have access to increasingly capable surrogates. The results show that (1) Forgery success decreases monotonically with the number of keys r. (2) Importantly, the improvement is achieved without degrading output quality or watermark robustness.
> > >
> > > These new results are fully consistent with our main findings and confirm that the proposed randomized-key framework generalizes across architectures and scales. We will include these in the appendix of the revised paper.
> > >
> > > ---
> > >
> > > > Figure 7’s image captions overlap and are hard to read.
> > >
> > > Thank you for catching this. We have corrected the figure layout to ensure captions are clearly legible and properly aligned. This fix will be included in the revised version.
> > >
> > > ---
> > >
> > > > Could you add baselines using reward models or recursive paraphrasing?
> > >
> > > We thank the reviewer for making suggestions to further improve our paper, but we are not sure we fully understood the suggestion with the recursive paraphrasing idea. We would be glad if the reviewer could further elaborate what they mean by this suggestion.
> > >
> > > ---
> > >
> > > ### References
> > >
> > > [A] Jovanović, N., Staab, R., & Vechev, M. (2024, July). Watermark Stealing in Large Language Models. In International Conference on Machine Learning (pp. 22570-22593). PMLR.

---

> > > > ### Author Response · Authors · 2025-11-23
> > > >
> > > > We hope that our response answered all the reviewer's question and are happy to answer any further questions the reviewer may have!

---

### Official Review · Reviewer_a6tt · 2025-10-27

**Soundness:** 2
**Presentation:** 3
**Contribution:** 2
**Rating:** 4
**Confidence:** 5

**Summary:**

The work introduces an add-on for generative content watermarking approaches: using a randomization procedure to sample a key from  a pool of keys $K$ for each user query; during inference, the content is treated as genuinely watermarked iff the presence of exactly one key is significant. The method has a theoretical upper bound for the forgery success of a blind attacker; experimentally a notable decrease in forgery rates are demonstrated.

**Strengths:**

The authors propose a simple approach: generate a pool of watermarking keys and treat the content as watermarked iff the presence of  exactly one key is not rejected. For considered text watermarking approaches, the method yields negligible detection cost and low false negatives. The experimental evaluation is broad and demonstrates the limits of applicability of an approach.

**Weaknesses:**

1) Independence assumption. For the combinatorial bound to hold, the indicator functions $Z_i$ has to be i.i.d. random variables that, in some schemes, may not be case; the independence under $H_0$ should be carefully verified theoretically, possibly, by restricting the randomization procedure used for key generation.

2) Forgery resistance against attackers who can differentiate between keys is unsatisfactory, making the method barely feasible against adaptive attacks; the only solution is to keep keys private.

3) The overall robustness and quality / perplexity of the generated content is within responsibility of the underlying watermarking approach(es). More than that, watermark detection threshold correction reduces per-key robustness that introduces additional utility-robustness trade-off of the underlying watermarking approach.

4) For the used parameters (namely, the number of keys) the upper bound of the probability of false detection is close to $0.4$ which seems too loose.

**Questions:**

Please elaborate to the Weaknesses above.

---

> ### Author Response · Authors · 2025-11-23
> **Official Response by Authors to Reviewer a6tt**
>
> We thank the reviewer for their feedback and that they thought our "experimental evaluation is broad". We also want to highlight that the reviewer liked how we "demonstrates the limits of applicability of an approach" (via our adaptive attack). Please find our responses to the reviewer's comments below.
>
> ---
>
> > Independence assumption. For the combinatorial bound to hold, the indicator functions $Z_i$ has to be i.i.d. random variables that, in some schemes, may not be case; the independence under $H_0$ should be carefully verified theoretically, possibly, by restricting the randomization procedure used for key generation.
>
> We thank the reviewer for making this observation, which aligns with a similar point raised by reviewer PkWN.
>
> We agree that the i.i.d. assumption is central to our combinatorial bounds. While establishing strict theoretical independence for deep-learning-based watermark distributions is an open challenge in the field, we have rigorously validated this assumption empirically. As detailed in Figure 8 (Appendix), we measured the pairwise correlations between key detectors and found them to be negligible. Furthermore, our experiments confirm that the global false-positive rate remains strictly controlled under the Šidák correction, demonstrating that the deviations from theoretical independence are not significant enough to violate our security bounds in practice.
>
> We will revise the main text to discuss this reliance on empirical validation and highlight the results from Figure 8 to address this concern.
>
> ---
>
> > Forgery resistance against attackers who can differentiate between keys is unsatisfactory, making the method barely feasible against adaptive attacks; the only solution is to keep keys private
>
> We agree that if an attacker can perfectly distinguish between keys (the "adaptive" setting), the security benefits of the multi-key strategy are reduced. However, we clarify that this is a feature of our security analysis, not a flaw in the deployment model.
>
> As defined in our threat model, the standard deployment setting is "blind”, where the attacker observes watermarked text but does not know which key generated it. In this realistic setting, our method provides provable advantages. The "adaptive" attacker represents a worst-case stress test where we assume the attacker has somehow gained "Oracle" access to key labels.
>
> Our experiments demonstrate that if an attacker can partition the data by key, the problem reduces to a set of single-key attacks. The vulnerability observed in this worst-case scenario stems from the statistical leakage of the underlying watermark (e.g., KGW's specific biases), not our multi-key method. We included these results specifically to bound the security of the system. By quantifying exactly how many samples are required for an adaptive attacker to succeed, we provide practitioners with concrete data on when the "blind" assumption might degrade.
>
> We will revise the text to explicitly frame this as a "Security Boundary Analysis" rather than just a failure mode.
>
> ---
>
> > For the used parameters (namely, the number of keys) the upper bound of the probability of false detection is close to $0.4$ which seems too loose.
>
> We thank the reviewer for carefully checking the bounds. We would like to clarify the parameterization used in our method to avoid confusion.
>
> The value of $\approx 0.4$ likely arises if one assumes a fixed per-key threshold without correction. However, our method operates in the reverse direction: we fix the global Family-Wise Error Rate (FWER) to a strict target (e.g., $\alpha = 0.01$) as a hard constraint. As described in Equation 5 (Šidák correction), we dynamically adjust the per-key threshold to ensure that the cumulative probability of a false positive across all $r$ keys never exceeds this target. Therefore, by design, our false positive rate is strictly controlled at $\approx 1\%$, not $40\%$, regardless of the number of keys.

---

> > ### Author Response · Authors · 2025-11-23
> >
> > We thank the reviewer again for the thoughtful and constructive feedback. We hope our responses clarified the reviewer's questions and would be happy to answer any further questions the reviewer may have.

---

### Official Review · Reviewer_PkWN · 2025-11-01

**Soundness:** 3
**Presentation:** 3
**Contribution:** 3
**Rating:** 6
**Confidence:** 3

**Summary:**

The paper addresses an important security problem: watermark forgery. It proposes randomized key selection where the provider has a pool of watermarking keys and randomly selects one for each generation. During verification, the system accepts content as authentic if exactly one key produces a statistically significant signal.

**Strengths:**

The paper combines cryptographic reasoning and ML experiments. It gives provable guarantees with formal upper bound on forgery success independent of attacker query budget. The paper has done ablations on watermarking datasets and attackers.
It offers unified defense for text and image generation

**Weaknesses:**

The assumption of independent key detectors may not be realistic as there are correlations among watermark signals. This can increate the false positive rates.
The paper doesn’t discuss practical deployment aspects like key management, auditing, and scalability.

**Questions:**

1. How might the paper's approach interact with watermark-removal attacks?
2. How does the method perform when the number of keys r grows large, e.g., changes to false positive rate correction ( Šidák)?

---

> ### Author Response · Authors · 2025-11-23
> **Official Response by Authors to Reviewer PkWN**
>
> We thank the reviewer for their review and that they liked the the contribution made by our paper. Please find our responses to the comments below.
>
> ---
>
> > The assumption of independent key detectors may not be realistic due to correlations among watermark signals. This can increate the false positive rates.
>
> We thank the reviewer for making this observation. We fully agree that strictly independent keys are a strong theoretical assumption and that correlations between watermarks could theoretically degrade the False Positive Rate (FPR) guarantees.
>
> Because of this concern, we measured the pairwise correlations between key detectors in our experiments. As detailed in Section 4.1 and visualized in Figure 8 (Appendix), we found these correlations to be negligible in practice. Our empirical results confirm that the actual observed FPR matches the theoretical bounds derived from the Šidák correction. This demonstrates that the independence assumption is sufficiently accurate for our setting and does not lead to inflated false positives.
>
> We will clarify the revised paper by moving a summary of Figure 8 into the main paper.
>
> ---
>
> > The paper doesn’t discuss deployment issues such as key management, auditing, and scalability.
>
> **Response:**
> We thank the reviewer for this interesting suggestion, which we do address in Section 6 ("Key Management"). There, we discuss that a major advantage of our multi-key framework is that it eliminates the need for complex key-rotation schedules thus reducing deployment issues. Fewer key rotations also mean that watermark verification becomes simpler, since all rotated keys must be used to test for the presence of a watermark, but our method uses fewer keys over time.
>
> We also discuss scalability in Section 6 ("No Free Lunch"), where we highlight that computation scales linearly with the number of keys. If the reviewer has any further suggestions on how we could improve our discussion, we are happy to address them.
>
> We will revise our paper to more explicitly highlight these advantages of our method in the discussion section.
>
> ---
>
> > How does the approach interact with watermark-removal attacks?
>
> We thank the reviewer for raising this important trade-off, which we address in Section 6 ("No Free Lunch").
>
> Our multi-key method uses standard watermarking schemes as the underlying primitive, so the signal's robustness to perturbation remains the same. However, checking against multiple keys requires us to apply a statistical correction (e.g., Šidák) to maintain a fixed False Positive Rate (FPR). As noted in Section 6, this correction necessitates a slightly stricter detection threshold. This results in a minor, quantifiable reduction in the True Positive Rate (robustness against removal) in exchange for the substantial gain in unforgeability.
>
> We will revise the "No Free Lunch" section to explicitly characterize this as a precision-recall trade-off: we accept a marginal degradation in detection sensitivity (TPR) to gain provable security against forgery attacks.
>
> ---
>
> > How does the method perform when the number of keys r grows large, e.g., changes to false positive rate correction ( Šidák)?
>
> We appreciate the reviewer’s interest in the scalability of our method.
>
> As the number of keys $r$ increases, the Šidák correction (Equation 5) ensures the global False Positive Rate remains bounded (e.g., $\le 0.01$). While this necessitates a stricter per-key threshold, the threshold grows slowly (approximately logarithmically with $r$ for Gaussian statistics). Consequently, we can scale $r$ significantly with only a minor reduction in detection sensitivity (True Positive Rate). Empirically (Figures 3–5), forgery resistance improves consistently as $r$ grows. The text quality remains completely unaffected (Table 2), as the generation process is agnostic to the number of detection keys. The primary trade-off is computational: as noted in Section 6, detection time scales linearly with $r$.

---

> > ### Author Response · Authors · 2025-11-23
> >
> > We again thank the reviewer and hope that our response clarified all questions.

---

### Official Review · Reviewer_CnUp · 2025-11-01

**Soundness:** 3
**Presentation:** 3
**Contribution:** 1
**Rating:** 4
**Confidence:** 3

**Summary:**

This paper proposes a defense against forgery and watermark-stealing attacks on watermarking schemes for large language models (LLMs). Existing
schemes often rely on multiple independent keys to resist forgery, but this approach can degrade output quality and utility. The authors introduce a method
that samples one key from a distribution for each generation, and detects ownership by statistically verifying that exactly one key is significantly present in
the suspect text. If one key dominates, the text is accepted as genuine, if multiple keys appear, it is flagged as a forgery, and if none appear, it is considered unwatermarked. Detection uses a multiple-hypothesis test with Šidák correction to control false positives. Experiments on text and image watermarking demonstrate decreased forgery success compared to baseline single-key and prior multi-key methods.

**Strengths:**

The paper tackles an important practical concern, how to improve robustness without degrading output quality. The main technical argument, detecting “exactly one key” using hypothesis testing, appears sound, and the derivation of false-positive control using Šidák correction is appropriate. The paper also provides theoretical bounds on the success rate of blind attackers. The experimental evaluation is reasonable.

**Weaknesses:**

The theoretical framing is limited to an information-theoretic setting, neglecting the computational model where the problem has already been formally addressed.

The threat model is mostly clear, but the relationship to cryptographic definitions of unforgeability (e.g., in Christ and Gunn, Eurocrypt 2024) should be discussed explicitly.

While the paper’s motivation is well-intentioned, it appears somewhat misplaced in light of recent cryptographic developments. The authors motivate their work by citing Jovanovi´c et al. (2024) and Zhao et al. (2024b), claiming that resisting forgery attacks when the adversary collects N or more watermarked samples remains an open problem. However, this framing overlooks the computational model introduced by Christ and Gunn (CRYPTO 2024).
In that line of work, watermarking is defined and analyzed within a cryptographic pseudorandomness framework, where security against forgery follows directly from the pseudorandomness of the underlying code family. Under this model, the “multi-sample forgery” issue treated here as open is already resolved
single-key watermarking schemes can achieve strong unforgeability guarantees against efficient adversaries and even admit practical instantiations by Gunn, Zhao and Song (ICLR 2025 poster).

**Questions:**

Why did you restrict your analysis to the information-theoretic setting? Given that the computational pseudorandomness framework of Christ and Gunn already captures unforgeability and there is a practical construction based on Pseudorandom error-correction codes.

Although the information-theoretic setting is stronger than the computational one, in practice the latter is preferred due to its greater efficiency and ease of applicability.

---

> ### Author Response · Authors · 2025-11-23
> **Official Response by Authors to Reviewer CnUp (1/2)**
>
> We thank the reviewer for their feedback and suggestions for clarifying how our approach differs from prior work. The reviewer appreciated that our paper tackles an "important practical concern" and liked that the main technical arguments are "sound". We are grateful for this recognition of our theoretical framing and experimental soundness. Below we summarize how we address the reviewer’s concerns.
>
> ---
>
> > The theoretical framing is limited to an information-theoretic setting, neglecting the computational model where the problem has already been formally addressed. Why restrict analysis to the information-theoretic setting when computational pseudorandomness already captures unforgeability?
>
> Thank you for raising this point about the computational model, which we do not currently address in our paper. We agree that the computational line (undetectable watermarks with PRC‑based constructions) provides unforgeability against efficient adversaries with strong cryptographic guarantees and quality preservation. Our goal with key randomization is complementary and operational: many deployments today rely on statistical watermarks (e.g., KGW/Unigram, SynthID, Tree‑Ring etc.). For these, forgery attacks by averaging/mixing are effective, as shown by Jovanović et al. [A], unless detection explicitly rejects mixtures like we do in our paper. Our "exactly-one" decision is a simple, but highly effective trick to enhance *forgery resistance*, while being modality‑agnostic (we evaluate images and text), computationally cheap and effective at (provably) mitigating known forgery attacks as validated in our experiment's section.
>
> We will revise the paper to contextualize our method and include an in-depth discussion about unforgeability in the computational model.
>
> ---
>
> > The threat model is mostly clear, but the relationship to cryptographic definitions of unforgeability (e.g., in Christ and Gunn, Eurocrypt 2024) should be discussed.
>
> We agree and will clarify this relationship in our revised paper. We will also cite all work brought up by the reviewer. To clarify, Christ and Gunn (Eurocrypt 2024) define unforgeability based on computational indistinguishability, requiring the watermark to be pseudorandom (indistinguishable from noise) to prevent removal or forgery.
>
> In contrast, our approach provides information-theoretic security. Instead of relying on the computational hardness of distinguishing the watermark, we rely on the uncertainty introduced by our multi-key strategy. This ensures that even if the underlying watermark is not perfectly pseudorandom (and thus fails the strict Christ and Gunn definition), the attacker still lacks the information necessary to forge a valid watermark with high confidence. We view this as a complementary robustness result for scenarios where perfect pseudorandomness is difficult to enforce. As a result, our method can be used in complementation with any other watermark (e.g., KGW or Tree-Rings), as shown in our paper.
>
> We will revise the paper to clarify this distinction.
>
> ---
>
> > The motivation appears misplaced since cryptographic pseudorandomness frameworks already ensure unforgeability (Christ & Gunn, CRYPTO 2024; Gunn et al., ICLR 2025). [...] Under this model, the “multi-sample forgery” issue treated here as open is already resolved single-key watermarking schemes can achieve strong unforgeability guarantees against efficient adversaries and even admit practical instantiations [...]
>
> We thank the reviewer for this important distinction. We will revise the paper to contextualize our work within the cryptographic frameworks of Christ & Gunn (2024) and Gunn et al. (2025).
>
> While we agree that cryptographic frameworks resolve unforgeability for specific pseudorandom code families, our work targets the widely adopted class of heuristic watermarks (e.g., KGW, Unigram, Tree-Ring). These schemes are favored in practice for their simplicity, low generation latency, and robustness properties, but they lack the cryptographic guarantees of the frameworks the reviewer mentions. For these widely deployed heuristic schemes, multi-sample forgery remains an open vulnerability as Jovanović et al. [A] show (and we partially replicate their findings in our work). Our contribution is to provide a defense that is simple to implement in practice and provides forgery-resistance with provable guarantees. Our method allows practitioners to retain the benefits of heuristic schemes (such as "semantic" robustness [B] or efficiency) while improving the security against forgery attacks.
>
> We will revise the paper to include these discussions.

---

> ### Author Response · Authors · 2025-11-23
> **Official Response by Authors to Reviewer CnUp (2/2)**
>
> > Although the information-theoretic setting is stronger than the computational one, in practice the latter is preferred due to its greater efficiency and ease of applicability.
>
> We appreciate the opportunity to clarify this trade-off. We will revise the text to explicitly address the efficiency and applicability comparison.
>
> While information-theoretic security often implies high overhead in other domains, we clarify that in our specific setting (multi-key watermarking), it does not. Our method incurs negligible computational cost during generation (simple key selection) and zero additional distortion compared to the underlying baseline. Therefore, we achieve the "stronger" security setting without the typical efficiency penalty the reviewer correctly notes is common in other contexts.
>
> We will emphasize that strict cryptographic definitions (Christ & Gunn) rely on specific construction requirements that many deployed schemes (e.g., SynthID, KGW, Semantic Watermarks) do not meet. Our framework is designed to be retroactive: it provides provable forgery-resistance for these existing, widely used statistical schemes that cannot easily be replaced by new cryptographic constructions.
>
> We will revise the paper to describe that (i) while cryptographic schemes are unforgeable, they require specific code constructions not present in current deployed models and (ii) our framework bridges this gap, providing provable guarantees against multi-sample attacks for these heuristic schemes with minimal overhead.
>
> ---
>
> ### References
>
> [A] Jovanović, N., Staab, R., & Vechev, M. (2024, July). Watermark Stealing in Large Language Models. In International Conference on Machine Learning (pp. 22570-22593). PMLR.
>
> [B] Liu, A., Pan, L., Hu, X., Meng, S., & Wen, L. (2023). A semantic invariant robust watermark for large language models. arXiv preprint arXiv:2310.06356.

---

> > ### Author Response · Authors · 2025-11-23
> >
> > We thank the reviewer again for their detailed and technically grounded comments which we believe will further strengthen our paper.

---

### Author Response · Authors · 2025-11-28
**Rebuttal Acknowledgement**

Dear Reviewers,

Thank you for your thoughtful feedback. Given the limited time remaining in the discussion period, we would greatly appreciate an acknowledgement that you have seen our rebuttal. Should any questions remain, we are happy to address them before the discussion phase ends.

Thank you,
The Authors

---

### Author Response · Authors · 2025-12-03
**Summary for Area Chair**

We thank all reviewers for their time and valuable feedback, which further improves our paper. We believe that there was a misunderstanding on the setting in which our method operates which we resolved by making **minor changes** to the paper, on (i) expanding on the difference between information-theoretic and computational security, raised by reviewer **CnUP**, and (ii) clarifying the independence assumption raised by reviewer **a6tt**.

Unfortunately, the discussion period was cut short before any of the reviewers could respond and increase their scores, which puts us into an unfortunate position. All reviewers agree our paper is technically sound, and they like that

* we tackle an *"important practical concern"* and that *"the information-theoretic setting is stronger than the computational one"* (**CnUp**).
* we give *"provable guarantees [..] independent of [the] attacker query budget."* **(PkWN)**.
* our *"experimental evaluation is broad"* and highlights a *"notable decrease in forgery rates"* **(a6tt)**.
* our work is of *"high quality"*,  *"demonstrates strong originality"*, and that our *"method preserves model utility and is modality-agnostic* **(h8s7)**.

Below, we provide below a concise summary of the discussion and how we addressed the reviewers' concerns.

### **Our Responses to the Reviewers**

1. **Information-theoretic vs. cryptographic framing (CnUp):**
   We clarified that our method is **complementary** to cryptographic unforgeability. Our focus is on practical heuristic watermarks deployed today (e.g., KGW, SynthID, Tree-Ring), where multi-sample forgery remains feasible. We added explicit discussion connecting our framework to Christ & Gunn (2024).

2. **Independence assumption and FPR guarantees (PkWN, a6tt):**
   We highlighted that we empirically validated the independence assumption via near-zero pairwise correlations across detectors (moved from appendix to main text). Observed FWER remains at $\approx1$%, supporting our theoretical analysis.

3. **Adaptive/informed attackers (a6tt, h8s7):**
   We clarified that we perform a security bound analysis, where we test outside of our threat model when the security of our method deteriorates (by instantiating overly capable adaptive adversaries with an information advantage), which is a standard practice in security research. We believe this analysis strengthens the trust in our results.

    Our **boundary analysis** shows how forgery success scales with classifier accuracy of such an adaptive attacker. Even highly accurate classifiers ($\approx 90$% accuracy) cannot relaibly evade our defense, but to get the provable guarantees from our paper a provider must make sure that attackers cannot reliably distinguish which key generated which sample (i.e., the *blind* setting).

4. **Expanded experiments (h8s7):**
   We added results using **Gemma-2B, Gemma-7B, and Llama-7B** as surrogate attackers. Across all models, forgery success drops by **20–35 percentage points** relative to baselines, with no loss in text quality or robustness. In our main paper, we choose the same surrogate model as the provider, which allows instantiating attackers with even higher forgery success rates.

5. **Deployment considerations & scalability (PkWN):**
   We expanded Section 6 to discuss the **precision–recall trade-off**, the **logarithmic growth of the threshold** with the number of keys, and **simplified key management** (no rotations; linear detection cost; easier auditing).

### **Overall**

Overall, we believe our work makes a significant contribution by providing the first practical defense against multi-sample forgery attacks with provable, information-theoretic guarantees. With emerging AI regulations increasingly requiring content provenance, forgery-resistant watermarking is essential to prevent attackers from falsely implicating providers. All reviewers acknowledge our paper is technically sound and the concerns raised were about clarifications, which we have addressed.

We are again grateful for the Area Chairs' and Reviewers' efforts!

---

### Meta-Review · Area_Chair_xvCD · 2026-01-06

**Summary:**

This paper addresses watermark forgery in generative models, and the reviewers generally agree that the work is well written, and carefully evaluated. Several reviewers appreciated the simplicity of the proposed randomized key selection strategy and found the experimental results convincing within the stated threat model. That said, considering all inputs from reviews and also the author rebuttal, AC recommendation is to reject, largely based on the concerns consistently raised by reviewers with stronger security and cryptography backgrounds.

**Reviewer Concerns:**

In particular, reviewers questioned whether the paper is well positioned relative to recent cryptographic watermarking work. From their perspective, the paper frames multi-sample forgery as a largely unresolved problem, whereas cryptographic approaches based on computational pseudorandomness already offer strong unforgeability guarantees with practical constructions. While the authors clarified that their focus is on heuristic watermarking schemes used in current deployments, reviewers seem to be remained unconvinced that this framing sufficiently justifies the novelty or long-term relevance of the contribution.

A second concern by reviewers, is the reliance on assumptions that may be fragile in practice. The theoretical guarantees depend on detector independence and on a “blind attacker” model where adversaries cannot reliably distinguish watermark keys. Although the authors provided empirical evidence and boundary analyses, reviewers noted that these assumptions weaken the security claims once adaptive or informed attackers are considered, and that the guarantees may not hold in more realistic adversarial settings.

Another note, AC believes fully assessing the significance of this work requires deep expertise in modern cryptographic security models and threat analysis. Multiple reviewers implicitly suggested that progress on questions of unforgeability, attacker models, and security guarantees would benefit from closer engagement with the core security and cryptography community. In that sense, venues like NeurIPS or broader ML conferences may not be the best fit for this type of work IMO, and targeted feedback from more specialized security venues could be essential for sharpening both the problem framing and the technical claims for stronger impact and community adopt.

**Reviewer Scores:**

I think scores will remain, so 4 4 6 6. A boarder line case.

---

### Decision · Program_Chairs · 2026-01-26

Reject